# Structural basis for COMPASS recognition of an H2B-ubiquitinated nucleosome

Evan J Worden, Xiangbin Zhang, Cynthia Wolberger*

Department of Biophysics and Biophysical Chemistry, Johns Hopkins University School of Medicine, Baltimore, United States

**Abstract** Methylation of histone H3K4 is a hallmark of actively transcribed genes that depends on mono-ubiquitination of histone H2B (H2B-Ub). H3K4 methylation in yeast is catalyzed by Set1, the methyltransferase subunit of COMPASS. We report here the cryo-EM structure of a six-protein core COMPASS subcomplex, which can methylate H3K4 and be stimulated by H2B-Ub, bound to a ubiquitinated nucleosome. Our structure shows that COMPASS spans the face of the nucleosome, recognizing ubiquitin on one face of the nucleosome and methylating H3 on the opposing face. As compared to the structure of the isolated core complex, Set1 undergoes multiple structural rearrangements to cement interactions with the nucleosome and with ubiquitin. The critical Set1 RxxxRR motif adopts a helix that mediates bridging contacts between the nucleosome, ubiquitin and COMPASS. The structure provides a framework for understanding mechanisms of trans-histone cross-talk and the dynamic role of H2B ubiquitination in stimulating histone methylation.

## Introduction

The histone proteins that package eukaryotic DNA into chromatin (*Andrews and Luger, 2011*) are subject to a huge variety of post-translational modifications that regulate chromatin structure, nucleosome positioning and protein recruitment, thereby playing a central role in regulating transcription (*Kouzarides, 2007*). Methylation of histone H3 at lysine 4 (H3K4) is a mark of actively transcribed genes and is enriched in promoter regions (*Barski et al., 2007*). H3K4 is methylated in yeast by the Set1 methyltransferase (*Roguev et al., 2001*), which can attach up to three methyl groups to the lysine ε-amino group (*Santos-Rosa et al., 2002*), and in humans by the six related SET1/MLL family of methyltransferases (*Meeks and Shilatifard, 2017*). Methylation of nucleosomal H3K4 depends on the prior ubiquitination of histone H2B (H2B-Ub) at Lysine 120 (Lys 123 in yeast) (*Dover et al., 2002*; *Shahbazian et al., 2005*; *Sun and Allis, 2002*), an example of histone modification 'cross-talk' in which attachment of one histone mark templates the deposition of another. H2B-Ub and H3K4 di- and tri-methylation are strongly associated with active transcription in both yeast and humans (*Barski et al., 2007*; *Jung et al., 2012*; *Liu et al., 2005*; *Minsky et al., 2008*; *Pokholok et al., 2005*; *Santos-Rosa et al., 2002*; *Shieh et al., 2011*; *Steger et al., 2008*). H3K4 methylation can serve as a recruitment signal for various transcription activators (*Sims et al., 2007*; *Vermeulen et al., 2010*; *Vermeulen et al., 2007*; *Wysocka et al., 2006*) including SAGA, whose acetyltransferase activity is stimulated by H3K4 methylation (*Bian et al., 2011*; *Ringel et al., 2015*).

In yeast, H3K4 is methylated by the evolutionarily conserved COMplex of Proteins ASsociated with Set1 (COMPASS) (*Miller et al., 2001*; *Shilatifard, 2012*), which contains Sgh1 (Cps15), Sdc1 (Cps25), Swd3 (Cps30), Swd2 (Cps35), Spp1 (Cps40), Swd1 (Cps50), Bre2 (Cps60), as well as Set1 (*Miller et al., 2001*; *Nagy et al., 2002*; *Roguev et al., 2001*; *Shilatifard, 2012*). Set1, which contains a conserved SET methyltransferase domain, is inactive on its own (*Avdic et al., 2011*; *Dou et al., 2006*; *Patel et al., 2009*; *Southall et al., 2009*). Multiple studies have identified a core subcomplex comprising Sdc1, Swd3, Swd1, Bre2 and Set1 which constitutes the minimal set of subunits required to support full Set1 methyltransferase activity (*Dou et al., 2006*; *Patel et al., 2009*; *Schneider et al.,*

*For correspondence:
cwolberg@jhmi.edu

*2005*; *Southall et al., 2009*). This core subcomplex of COMPASS is highly conserved across all eukaryotes, with human WDR5/RbBP5/ASH2L/DPY30 corresponding to yeast Swd3/Swd1/Bre2/Sdc1 (*Miller et al., 2001*; *Shilatifard, 2012*). The ability of COMPASS activity to be fully stimulated on nucleosomes by H2B ubiquitination depends upon the presence of an additional COMPASS subunit, Spp1, as well as the RxxxRR motif in the N-set region of Set1 which is also referred to as the Arginine Rich Motif (ARM) (*Hsu et al., 2019*; *Kim et al., 2013*). We refer to this core subcomplex plus Spp1 as the H2B-ubiquitin-sensing subcomplex.

Structural studies of the yeast core subcomplex (*Hsu et al., 2018*) and the H2B-ubiquitin-sensing subcomplex (*Qu et al., 2018*; *Takahashi et al., 2011*) have shown that COMPASS adopts a Y-shaped, highly intertwined structure with the Set1 catalytic domain at its core. Furthermore, a recent structure of the related human MLL1 core complex bound to a ubiquitinated nucleosome revealed the underlying mechanisms of nucleosome recognition by human COMPASS-like complexes (*Xue et al., 2019*). In addition, a recent structure of the related COMPASS complex from *K. Lactis* has shown how COMPASS binds to a ubiquitinated nucleosome (*Hsu et al., 2019*). However, there is currently no structural information on how the full H2B-ubiquitin-sensing COMPASS subcomplex from *Saccharomyces cerevisiae* binds and recognizes the H2B-Ub containing nucleosome. We report here the 3.37 Å resolution cryo-EM structure of the H2B-Ub sensing COMPASS subcomplex from the yeast, *Saccharomyces cerevisiae*, bound to an H2B-Ub nucleosome. The structure shows that COMPASS contains multiple structural elements that position the complex on the nucleosome disk though interactions with nucleosomal DNA and three of the core histones. The position of the Set1 catalytic domain suggests that COMPASS methylates H3K4 in an asymmetric manner by targeting H2B-Ub and H3K4 on opposite sides of the nucleosome. Structuring of a critical RxxxRR motif to form a helix enables the complex to associate with the nucleosome acidic patch, with the RxxxRR helix forming the bottom edge of an extended ubiquitin interaction crevice that underlies the structural basis of H2B-ubiquitin recognition by COMPASS. Comparison with other ubiquitin-activated methyltransferases shows that interactions with the H2B-linked ubiquitin are highly plastic and suggests how a single ubiquitin mark can be utilized by several different enzymes. Our findings shed light on the long-standing mystery of how H2B-Ub is recognized by COMPASS and provide the first example of trans-nucleosome histone crosstalk.

## Results

### Architecture of the COMPASS H2B-Ub nucleosome complex

We determined the cryo-EM structure of the minimal, H2B-ubiquitin-sensing subcomplex of *Saccharomyces cerevisiae* COMPASS bound to a *Xenopus laevis* nucleosome core particle ubiquitinated at histone H2B K120 via a non-hydrolyzable dichloroacetone (DCA) linkage (*Morgan et al., 2016*). The ubiquitinated residue corresponds to K123 of yeast H2B. To drive tight association between COMPASS and the nucleosome, we utilized a variant of histone H3 in which K4 was substituted with the non-native amino acid, norleucine (Nle) (*Worden et al., 2019*). Lysine-to-norleucine mutations have been shown to greatly increase the affinity of SET-domain methyltransferases for their substrates in a S-adenosylmethionine (SAM)-dependent manner (*Jayaram et al., 2016*; *Lewis et al., 2013*; *Worden et al., 2019*). To assess the gain in affinity imparted by the H3K4Nle substitution, we used gel mobility shift assays to measure binding of COMPASS to different nucleosome variants in the presence of SAM (*Figure 1—figure supplement 1*). Surprisingly, COMPASS binds to unmodified and H2B-Ub nucleosomes with the same apparent affinity, indicating that H2B-Ub does not contribute significantly to the energy of COMPASS binding to the nucleosome (*Figure 1—figure supplement 1*). However, H2B-Ub nucleosomes that also contain the H3K4Nle mutant bind COMPASS with 2–5 fold higher affinity than unmodified nucleosomes (*Figure 1—figure supplement 1*, compare the 0.125 µM lane for all samples). Methyltransferase activity assays on an H3 peptide fragment (residues 1–21) confirmed that the COMPASS was active (*Figure 1—figure supplement 1*). We therefore prepared complexes between COMPASS and H2B-Ub nucleosomes containing the H3K4Nle mutation in the presence of saturating SAM and determined the structure of the complex to 3.37 Å by single particle cryo-EM (*Figure 1a*, *Figure 1—figure supplements 2–3* and *Table 1*).

In the reconstruction, two COMPASS complexes are bound to opposite faces of the nucleosome in a pseudo-symmetric 2:1 arrangement (*Figure 1a*, *Figure 1—figure supplement 2*). However, only

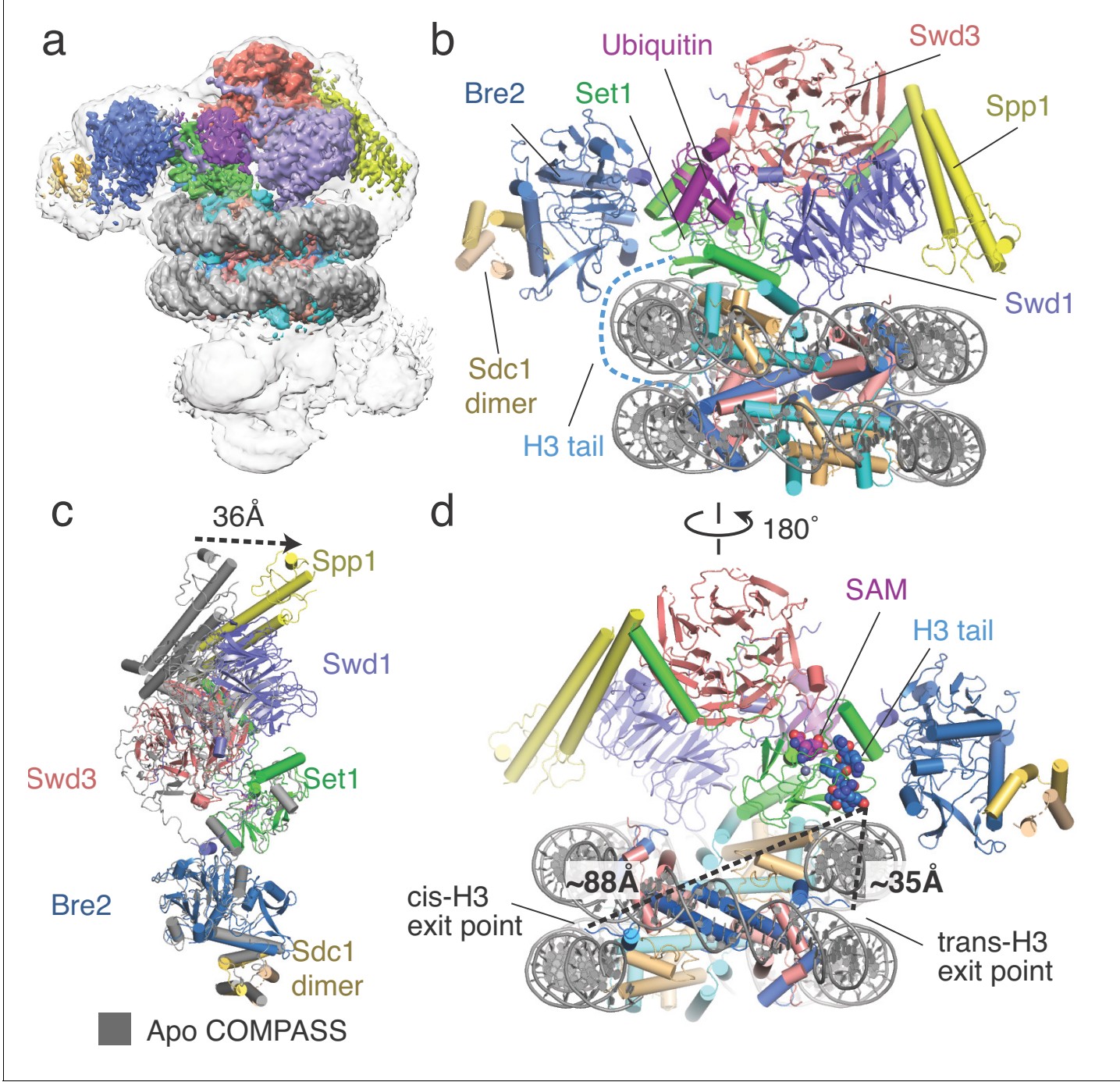

**Figure 1.** Architecture of the COMPASS H2B-Ub nucleosome complex. (a) Cryo-EM structure of the COMPASS-nucleosome complex. The unsharpened EM density showing two COMPASS molecules bound to the nucleosome is depicted as a semitransparent surface. The sharpened EM density of the complex is depicted as an opaque surface and colored according to the different subunits of the complex. (b) The model of the COMPASS-nucleosome complex is shown and colored as in panel a. The unstructured histone H3 tail residues between the H3 exit point in the nucleosome and the Set1 active site are depicted as a blue dashed line. (c) Large-scale structural motions of COMPASS from the nucleosome-free state to the nucleosome-bound state. Free COMPASS (PDB: 6BX3) is colored gray and nucleosome-bound COMPASS is colored according to panel b. The largest motion in the structural transition at the end of Spp1 is shown as a black dashed arrow. (d) The COMPASS-nucleosome structure viewed from the dyad axis. The distances between the cis-H3 and trans-H3 subunits and the H3 residues in the Set1 active site are shown as dashed black lines. The online version of this article includes the following figure supplement(s) for figure 1:

**Figure supplement 1.** Analysis of COMPASS activity and binding to the nucleosome.

**Figure supplement 2.** Cryo-EM processing pipeline.

*Figure 1 continued on next page*

*Figure 1 continued*

**Figure supplement 3.** Cryo-EM map and model validation and example density.
**Figure supplement 4.** Detailed views of EM density in the structure.

one of the two bound COMPASS assemblies resolved to high resolution. The final model, therefore, includes one COMPASS complex and the nucleosome core particle (*Figure 1a–b*). To build the yeast COMPASS complex, models of Spp1 and the N-set region of Set1 were taken from the cryo-EM structure of *S. cerevisiae* COMPASS (*Qu et al., 2018*) and docked into the EM density. For the rest of the COMPASS model, crystal structures of *K. lactis* Bre2, Swd1, Swd3, Sdc1 and Set1 subunits (*Hsu et al., 2018*) were utilized to create homology models with the *S. cerevisiae* sequence (25%–50% sequence identity) using Swiss-model (*Waterhouse et al., 2018*). The homology models were docked into the EM density, manually re-built in COOT (*Emsley et al., 2010*) and refined using

**Table 1.** Cryo-EM data collection, refinement and validation statistics

| | Complex between COMPASS and the H2B-Ub nucleosome (EMDB-21157) (PDB 6VEN) |
|---|---|
| Data collection and processing | |
| Magnification | 81,000 |
| Voltage (kV) | 300 |
| Electron exposure (e–/Å$^2$) | 50 |
| Defocus range (µm) | −1.0 to −2.5 |
| Pixel size (Å) | 1.08 |
| Symmetry imposed | C1 |
| Initial particle images (no.) | 2,036,654 |
| Final particle images (no.) | 179,588 |
| Map resolution (Å) FSC threshold | 3.37 (0.143) |
| Map resolution range (Å) | 999–3.37 |
| Refinement | |
| Initial model used (PDB code) | PDB: 6NJ9, 6BX3, 6CHG |
| Model resolution (Å) FSC threshold | 3.40 (0.5) |
| Model resolution range (Å) | 47.6 to 3.40 |
| Map sharpening *B* factor (Å$^2$) | −122.6 |
| Model composition Non-hydrogen atoms Protein residues Ligands | 24,032 2271 2 |
| *B* factors (Å$^2$) Protein Ligand | 102.57 113.31 |
| R.m.s. deviations Bond lengths (Å) Bond angles (°) | 0.004 0.696 |
| Validation MolProbity score Clashscore Poor rotamers (%) | 1.92 9.79 0.05 |
| Ramachandran plot Favored (%) Allowed (%) Disallowed (%) | 94.01 5.99 0.0 |

Phenix (*Adams et al., 2010*) (see Materials and methods). Spp1, Bre2 and the Sdc1 dimer are less well resolved than the other COMPASS subunits due to their location on the periphery of the complex and consequent higher mobility (*Figure 1*, *Figure 1—figure supplement 3*). In particular, the EM density corresponding to the N-terminal portion of Spp1 was very weak and precluded accurate model fitting. Therefore, the N-terminal portion of Spp1 was excluded from our final model (*Figure 1—figure supplement 4*).

The COMPASS complex spans the entire diameter of the nucleosome and is anchored by contacts between DNA and Bre2/Set1 at one end of the complex and Swd1 and Spp1 at the other (*Figure 1b*, *Figure 2*). These DNA contacts position COMPASS such that Swd1 and the Set1 catalytic domain can contact the central histone core. Compared to isolated *S. cerevisiae* COMPASS (*Qu et al., 2018*), nucleosome-bound COMPASS flexes around the central Set1 catalytic domain and

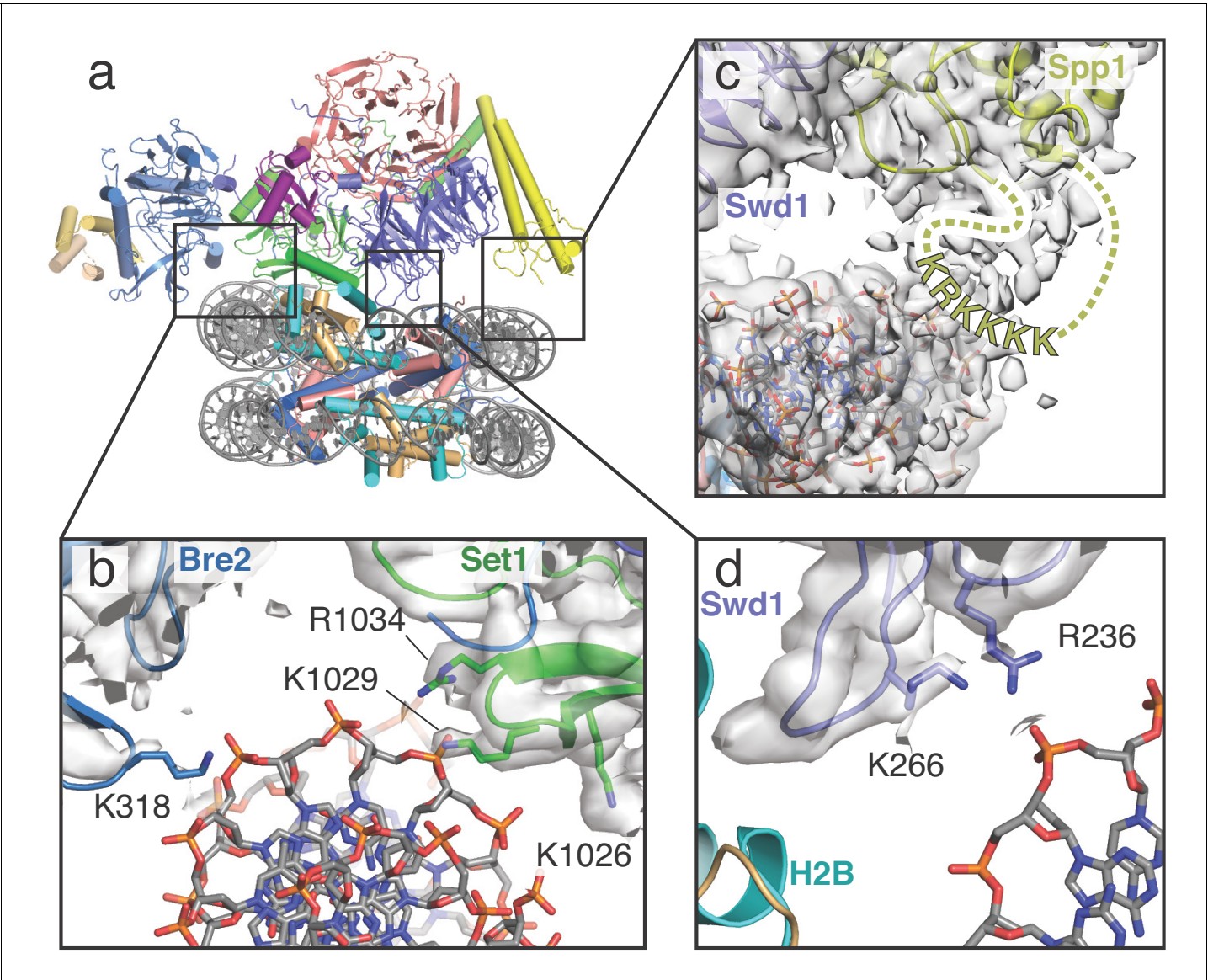

**Figure 2.** COMPASS makes multiple contacts with nucleosome DNA. (**a**) Model of the COMPASS-nucleosome structure shown in cartoon representation and colored as in *Figure 1*. (**b–d**) Detailed views of COMPASS interactions with nucleosomal DNA. In all views the experimental EM density is shown as a semi-transparent gray surface. (**b**) DNA contact between Set1 and Bre2. Potential DNA-interacting COMPASS residues are shown in stick representation. (**c**) Contact between a flexible loop in Spp1 and the nucleosomal DNA. The unmodeled loop of Spp1 is depicted as a yellow dashed line and the positively charged, putative DNA-interacting residues in the loop are shown as yellow letters. (**d**) Interaction between Swd1 and the nucleosome DNA. Swd1 residues that contact DNA are shown in stick representation.

moves Swd3, Spp1 and Swd1 toward the nucleosome by ~36 Å (*Figure 1c*). This movement allows Spp1 to bind the nucleosomal DNA, and allows Swd1 to interact with the histone core. Notably, a subset of particles in the cryo-EM structure of COMPASS in the absence of nucleosome (*Qu et al., 2018*) exhibited flexing about the same axis, although not to the extent observed here when COMPASS is bound to a nucleosome (*Figure 1c*). Our structure suggests that the previously observed conformational flexibility of COMPASS is important for nucleosome recognition.

The Set1 active site is oriented away from the nucleosome and contains density for the SAM cofactor and the bound H3 tail (*Figure 1d*, *Figure 1—figure supplement 3*). The intervening sequence of the H3 tail, from its exit point in the nucleosome (P38) to the first resolved residue in the Set1 active site (R8), is not visible in the maps, suggesting that these residues are highly mobile (*Figure 1b,d*). To determine which copy of histone H3 was connected to the portion of the H3 tail bound to Set1, we compared the distance between the last H3 residue in the Set1 active site (R8) and the first H3 residue (P38) on each face of the nucleosome. The Set1 active site is positioned much closer to the exit point of the H3 subunit on the opposite face of the nucleosome (trans-H3,~35 Å) than to the exit point of the H3 subunit on the same side of the nucleosome (cis-H3,~88 Å, *Figure 1d*). The 88 Å end-to-end distance between the exit point of cis-H3 and H3 in the Set1 active site is too long to be spanned by the intervening unstructured H3 tail residues given that an even greater distance (~100 Å) would be needed for the H3 residues to wrap around the nucleosomal DNA near the dyad axis. This arrangement therefore indicates that COMPASS methylates the nucleosome in an asymmetric manner by recognizing ubiquitin on one face of the nucleosome and targeting H3K4 on the opposite, trans-H3, face of the nucleosome.

COMPASS also interacts directly with the core histone proteins. A pair of loops in Swd1 anchor COMPASS near the C-terminal H2B helix (*Figure 3*). In addition, a long helix in Set1 containing the RxxxRR motif (*Kim et al., 2013*) extends along the surface of the histone core and makes multiple interactions with the nucleosome acidic patch formed by histones H2A and H2B (*Figure 4*). Finally, Set1, Bre2 and Swd1 bind to the H2B-linked ubiquitin (*Figures 5* and *6*), providing a structural basis for the crosstalk between H2B-Ubiquitination and H3K4 methylation.

## COMPASS interacts with DNA using three distinct interfaces

COMPASS interacts with DNA in three distinct locations, thereby orienting the complex on the face of the nucleosome (*Figure 2*). At one end of COMPASS, the nucleosomal DNA docks into a concave surface at the interface of Bre2 and Set1 (*Figure 2a,b*). This concave surface is lined with several basic residues that can potentially contact the DNA. In particular, Bre2 K318 is in a position to interact directly with the sugar-phosphate backbone (*Figure 2b*). We note that we did not observe clear sidechain density for Bre2 K318, so the position of this sidechain is inferred from the conformation of the protein backbone. Set1 residues R1034, K1029 and K1026 are also located at this interface (*Figure 2b*) but are too far away to directly contact the DNA. Instead, these basic residues likely serve to increase the local positive charge of the concave DNA binding surface. On the opposing edge of the nucleosome, COMPASS contacts the DNA at two distinct interfaces mediated by Spp1 and Swd1 (*Figure 2c,d*). Fragmented density connecting Spp1 and the nucleosomal DNA corresponds to a loop in Spp1 that is disordered in our structure (residues 241–261). This Spp1 loop (*Figure 2c*) contains a patch of positively-charged amino acids (KRKKKK) that likely interact with the negatively charged DNA backbone and major groove. Swd1 interacts with the nucleosomal DNA using two highly conserved basic resides, R236 and K266, which emanate from loops in blade 5 of the Swd1 WD40 domain (*Figure 2d*). Substitution of Swd1 K266 with an alanine decreased H3K4 di- and tri-methylation in yeast (*Figure 3c–d*), indicating that the loss of even a single DNA contact can impair COMPASS function. As discussed further below, this region of Swd1 is highly conserved and also mediates contacts between Swd1 and the core histone octamer, making this part of Swd1 a highly utilized surface for COMPASS interaction with the nucleosome.

Importantly, similar interactions between these COMPASS subunits (Bre2, Spp1 and Swd1) and nucleosomal DNA have recently been observed in the human MLL1 complex (*Xue et al., 2019*) and the *K. lactis* COMPASS complex (*Hsu et al., 2019*), indicating that these DNA interactions are critical for COMPASS function and are highly conserved.

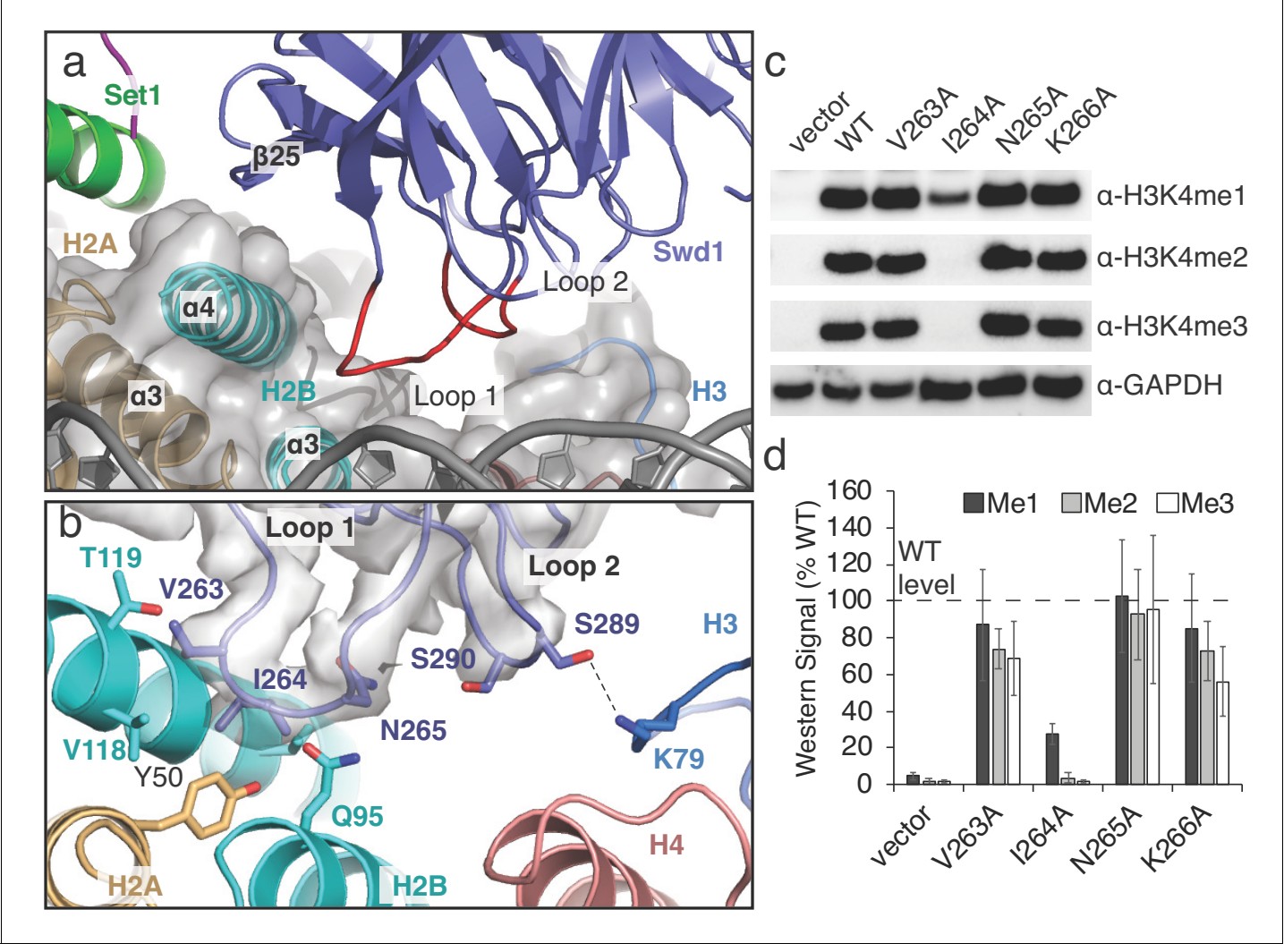

**Figure 3.** A conserved loop in Swd1 contacts the histone core. (a) Close up view of the contact between Swd1 and the histone core. Swd1 is shown as a blue cartoon and the two loops that contact the histone core are colored red. (b) Detailed view of the specific contacts made between Loop 1 and Loop 2 of Swd1 and the histone core. Potential hydrogen bonds are shown as black dashed lines and the sharpened EM density is shown as a transparent gray surface. (c) Western blot analysis of H3K4 methylation states in swd1Δ yeast strains transformed with plasmids containing the indicated swd1 variants. (d) Quantification of the western blots from panel c. Error bars are the standard deviation of the data (n = 4).

The online version of this article includes the following source data and figure supplement(s) for figure 3:

**Source data 1.** Western blot quantification for Swd1 Loop 1 mutants.
**Figure supplement 1.** Multiple sequence alignments of Swd1, Set1 and Bre2.

## A conserved loop in Swd1 contacts the core histone octamer

The structure reveals that Swd1 contains two loops that interact with three different histones in the nucleosome core (*Figure 3a*). Swd1 Loop 1 connects β21 and β22, and, along with the edge of β-strand 25, embraces the H2B C-terminal helix, α4 (*Figure 3a*). At the tip of L, V263 and I264 insert into a small hydrophobic crevice at the three-helix interface consisting of H2B helices, α3 and α4, and H2A α3 (*Figure 3a,b*). This hydrophobic crevice includes H2A Y50, and H2B V118, which are in van der Waals contact with Swd1 V263 and I264. In addition to the hydrophobic interactions in L, Swd1 N265 is positioned close to H2B Q95, potentially forming a hydrogen bond and further stabilizing the Loop 1 interaction with the histone core (*Figure 3b*). To assess the importance of the interaction between Swd1 Loop 1 and the H2A/H2B hydrophobic crevice, we examined the effects of alanine substitutions on histone H3K4 methylation in *S. cerevisiae*. As shown in *Figure 3c*, the

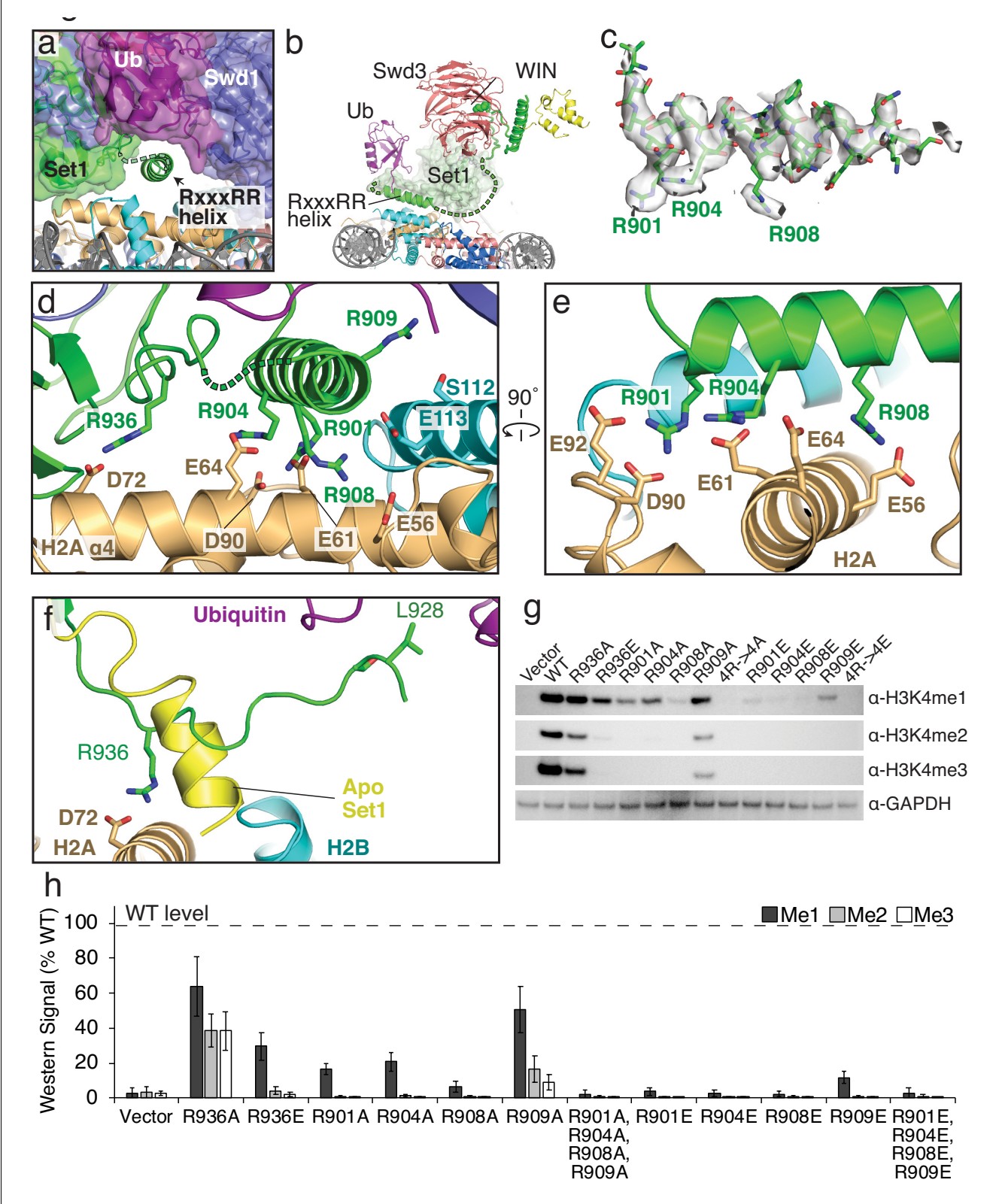

**Figure 4.** The RxxxRR helix binds to the H2A/H2B acidic patch. (a) The COMPASS-nucleosome structure with COMPASS subunits surrounded by semi-transparent colored surfaces. The RxxxRR helix passes underneath the H2B-linked ubiquitin and is shown in cartoon representation. (b) Cutaway view of the COMPASS nucleosome structure showing the RxxxRR helix passing between COMPASS and the nucleosome. (c) Sharpened EM density surrounding the RxxxRR helix, which is shown in stick representation. (d–e) End-on view (d) and side view (e) of the RxxxRR helix. Contacting residues

*Figure 4 continued on next page*

*Figure 4 continued*

between the RxxxRR helix and the H2A/H2B acidic patch are shown in stick representation. (**f**) Detailed view of the section of Set1 that undergoes a conformational change. The corresponding part of Set1 in the isolated COMPASS structure (PBD: 6BX3, Apo) is colored yellow and shown as a cartoon. The Set1 residues that make contacts with the histone core and Ubiquitin are shown in stick representation and colored green. (**g**) Western blot analysis of H3K4 methylation states in *set1Δ* yeast strains transformed with plasmids containing the indicated *set1* variants. (**h**) Quantification of the western blots from panel **g**. Error bars are the standard deviation of the data (n = 3).

The online version of this article includes the following source data for figure 4:

**Source data 1.** Western blot quantification for Set1 RxxxRR helix mutants.

substitution with the greatest effect was Swd1 I264A, which completely abolished H3K4 di- and tri-methylation and greatly reduced H3K4 mono-methylation (*Figure 3c,d*). I264 is positioned in the center of the H2A/H2B hydrophobic crevice and the strong defect in H3K4 methylation seen for the I264A mutation indicates that this interaction is critical for COMPASS activity in vivo. The V263A mutant slightly decreased H3K4 di- and tri-methylation, but did not change H3K4 mono-methylation, whereas an N265A mutation had no effect (*Figure 3c,d*). We note that it is not possible to determine from these data if the changes in the relative H3K4 mono-, di- and tri-methylation levels in the COMPASS mutants are caused by altered product specificity, or slower enzyme turnover. Loop 1 is highly conserved (*Figure 3—figure supplement 1*) and the residues that correspond to V263 and I264 are always hydrophobic in character, indicating that that the interaction we observe between Loop 1 and the histone octamer is structurally conserved among Swd1 homologs. Indeed, the human MLL1 core complex subunit, Rbbp5, binds to the histone octamer using a similar interface (*Xue et al., 2019*) and another recent study reports that the *K. lactis* COMPASS complex Swd1 subunit also utilizes the hydrophobic cleft interface in a similar manner (*Hsu et al., 2019*).

In addition to Swd1 L, Loop 2 connects β-strands β23 and β24 and is oriented toward the histone H3 α2-α3 loop and histone H4 α3 (*Figure 3b*). At the tip of Loop 2, S289 forms a hydrogen bond with H3 K79. As compared to L, Loop 2 is not well conserved (*Figure 3—figure supplement 1*) and the contact it makes with H3K79 is not recapitulated in other COMPASS-like complexes (*Xue et al., 2019*). Taken together, these results show that the interaction between Swd1 and the nucleosome is conserved from yeast to humans and is critical for COMPASS function in vivo.

## The Set1 RxxxRR motif forms an adaptor helix that bridges the nucleosome acidic patch and ubiquitin

The arginine-rich RxxxRR motif in the N-set region of Set1 is critical for stimulation of methyltransferase activity by H2B ubiquitination and mutants in this motif impair COMPASS activity in yeast (*Kim et al., 2013*). Our structure reveals the critical role that the RxxxRR motif plays in COMPASS interactions with both the nucleosome and ubiquitin. In isolated yeast COMPASS (*Qu et al., 2018*), the RxxxRR motif is disordered. When bound to the H2B-Ub nucleosome, Set1 residues 897–921 become ordered, forming a long helix that passes underneath COMPASS and makes extensive contacts with the nucleosome acidic patch (*Figure 4a–e*), as well as with the H2B-linked ubiquitin (see below). The RxxxRR helix docks on the nucleosome surface parallel to the C-terminal helix α4 of histone H2B, positioning multiple arginine residues opposite the negatively charged residues of H2A that make up the acidic patch (*Figure 4d,e* and *Figure 1—figure supplement 4*). Residues in the Set1 RxxxRR helix form several specific electrostatic interactions with acidic patch residues: R908 is in a position to contact H2A residue E56 and H2B E113, R904 contacts H2A residue E61 and E64 and R901 contacts H2A residues D90 and E92 (*Figure 4d,e*). The extensive electrostatic network mediated by the RxxxRR helix is flanked by interactions between Set1 R936 and H2A D72 on one side and between R909 and H2B S112 on the other. We note, however, that there is no clear side-chain density for R909 in our structure, so the contact between R909 and S112 is inferred from the conformation of the backbone. Interestingly, the RxxxRR motif is conserved in human Set1 orthologs, but not in the human paralogs MLL1-4 (*Figure 3—figure supplement 1*) which have recently been shown not to bind the nucleosome acidic patch (*Xue et al., 2019*).

In addition to formation of the RxxxRR helix, nucleosome binding is accompanied by a profound restructuring of Set1 α-helix, 926–933, which unravels and forms an extended strand that lies parallel to the RxxxRR helix (*Figure 4f* and *Figure 1—figure supplement 4*). This extended strand lies along the histone core, orienting R936 toward the nucleosome surface (*Figure 4d,f*). The conformational

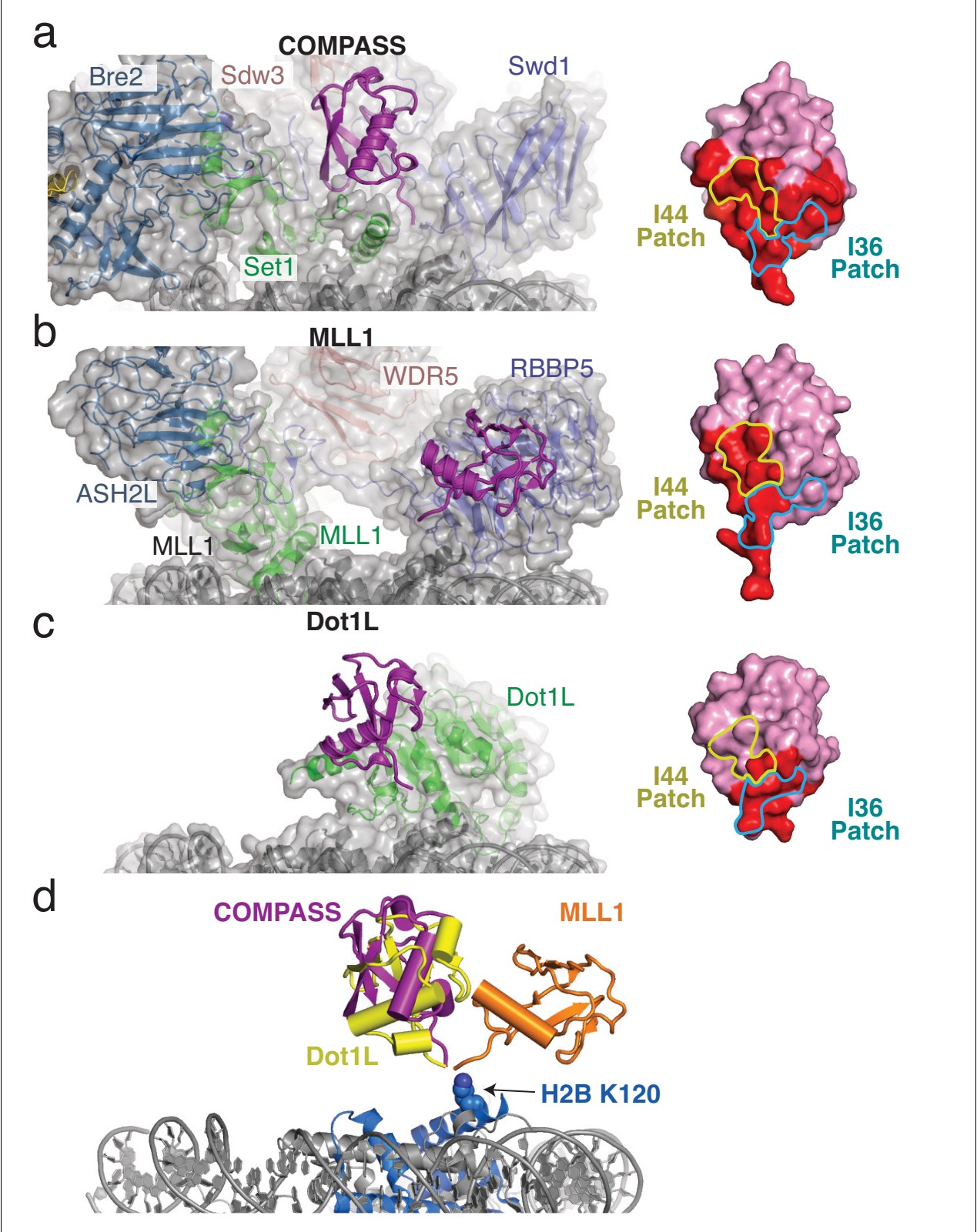

**Figure 5.** H2B-Ubiquitin recognition by different methyltransferases. (**a–c**) Structures of H2B-ubiquitin activated methyltransferases bound to the H2B-Ub nucleosome. The structures are encompassed by a semitransparent gray surface, except for ubiquitin which is colored purple and depicted as a cartoon. Surface representations of the bound ubiquitin are shown on the right with the buried surface area colored red. The locations of the I44 and I36 patches are indicated with colored yellow and blue lines, respectively. (**a**) Structure of COMPASS bound the H2B-Ub nucleosome. (**b**) Structure of

*Figure 5 continued on next page*

Figure 5 continued

the MLL1 core complex bound to the H2B-Ub nucleosome (PDB: 6KIU). (c) Structure of Dot1L bound to the H2B-Ub nucleosome (PDB: 6NJ9). (d) Superimposition of the structures from a–c but only showing the ubiquitin to compare the relative positions of the H2B-linked Ubiquitin. H2B K120 is shown as spheres and colored blue.

change in this region of Set1 alleviates what would otherwise be steric clash between the N-terminus of the helix formed by these residues in uncomplexed (Apo) COMPASS.

Previous mutagenesis studies of the N-set region of yeast Set1 (*Kim et al., 2013*) showed that a combined R909, R908 and R904 triple mutant abolished COMPASS activity *in vitro* and *in vivo*. Furthermore, a large-scale alanine screen of histone residues in yeast previously determined that residues in the H2A/H2B acidic patch are required for H3K4 methylation by COMPASS (*Nakanishi et al., 2008*). To assess the contribution of individual Set1 amino acids to H3K4 methylation by COMPASS *in vivo*, we generated yeast strains in which a *set*1 deletion was complemented with mutant *set*1 containing point mutations designed to disrupt interactions with the nucleosome acidic patch. As compared to wild-type, R936A and R936E Set1 mutations both reduced H3K4 methylation (*Figure 4g,h*). The charge reversal mutation, R936E, was more severe and almost entirely abolished di- and tri-methylation by COMPASS (*Figure 4g,h*). Individual alanine substitutions of R901A, R904A and R901A reduced mono-methylation by COMPASS and abolished di-, and tri-methylation. The R909A mutation, which does not form electrostatic contacts with the acidic patch, greatly reduced di-and tri-methylation. As expected, the quadruple mutant R901A, R904A, R908A and R909A (4R->4A) completely abolished all methylation of H3K4. Charge reversal mutants R901E, R904E, R908E, R909E and the quadruple 4R->4E mutants all abolished H3K4 methylation, expect for R909E, which retained some residual H3K4 mono-methylation (*Figure 4g,h*). Together, these structural and mutational data indicate that interactions between the RxxxRR helix and the acidic patch are critical for COMPASS activity.

## Structural basis of ubiquitin recognition

The COMPASS-Ubiquitin interaction is different from, and much more extensive than, the interaction between ubiquitin and the MLL1 core complex subunit RbBP5 (*Xue et al., 2019*) or Dot1L (*Anderson et al., 2019*; *Valencia-Sánchez et al., 2019*; *Worden et al., 2019*), the histone H3K79 methyltransferase which is also stimulated by H2B-Ub (*Briggs et al., 2002*; *McGinty et al., 2008*; *Ng et al., 2002*) (*Figure 5a–c*). Moreover, the H2B-linked ubiquitin is positioned in different orientations relative to the face of the nucleosome in each of these complexes (*Figure 5d*), indicating that H2B-Ub is a conformationally plastic epitope that can be recognized in structurally distinct ways. Ubiquitin binds to COMPASS in a large cleft located between Swd1, Set1 and Bre2 (*Figure 6a*), burying 930 Å$^2$ of total surface area. The H2B-linked ubiquitin sits on top of the Set1 RxxxRR helix and makes multiple contacts with the N-terminal and C-terminal extensions of Swd1 (*Figure 6a,b*). In addition to the Set1 and Swd1 contacts, there is substantial connecting density between Bre2 and the N-terminus of ubiquitin that likely corresponds to a 38 amino acid loop in Bre2 (residues 140–178) that is unmodeled in our structure (*Figure 6a*). This Bre2 loop is not found in human or other yeast COMPASS complexes and appears to be specific to the *S. cerevisiae* COMPASS complex (*Figure 3—figure supplement 1*).

The primary contact between ubiquitin and COMPASS is mediated by the N- and C-terminal extensions of Swd1 (*Figure 6b*). These Swd1 extensions are critical for COMPASS assembly and mediate multiple contacts between COMPASS subunits (*Hsu et al., 2018*; *Qu et al., 2018*). Compared to the nucleosome-free (apo) structure of COMPASS (*Qu et al., 2018*), the N- and C- terminal extensions of Swd1 change conformation when bound to ubiquitin (*Figure 6c,d*). These Swd1 extensions present an extended surface of hydrophobic residues which interact with the ubiquitin C-terminal tail and the hydrophobic 'I44 patch' on ubiquitin comprising I44, V70, L8 and H68 (*Figures 5a* and *6b*). The I44 patch is a canonical interaction surface that is contacted by many different ubiquitin binding proteins (*Komander and Rape, 2012*), and by the related MLL1 complex subunit, Rbbp5 (*Xue et al., 2019*) (*Figure 5b*). At the center of the I44 patch interaction, Swd1 L12, V401 and P8 contact Ub I44 and Swd1 F9 interacts with Ub L8, V70, H68 and the aliphatic portion of K6 (*Figure 6b,e* and *Figure 1—figure supplement 4*). The I44 patch contact is flanked by electrostatic

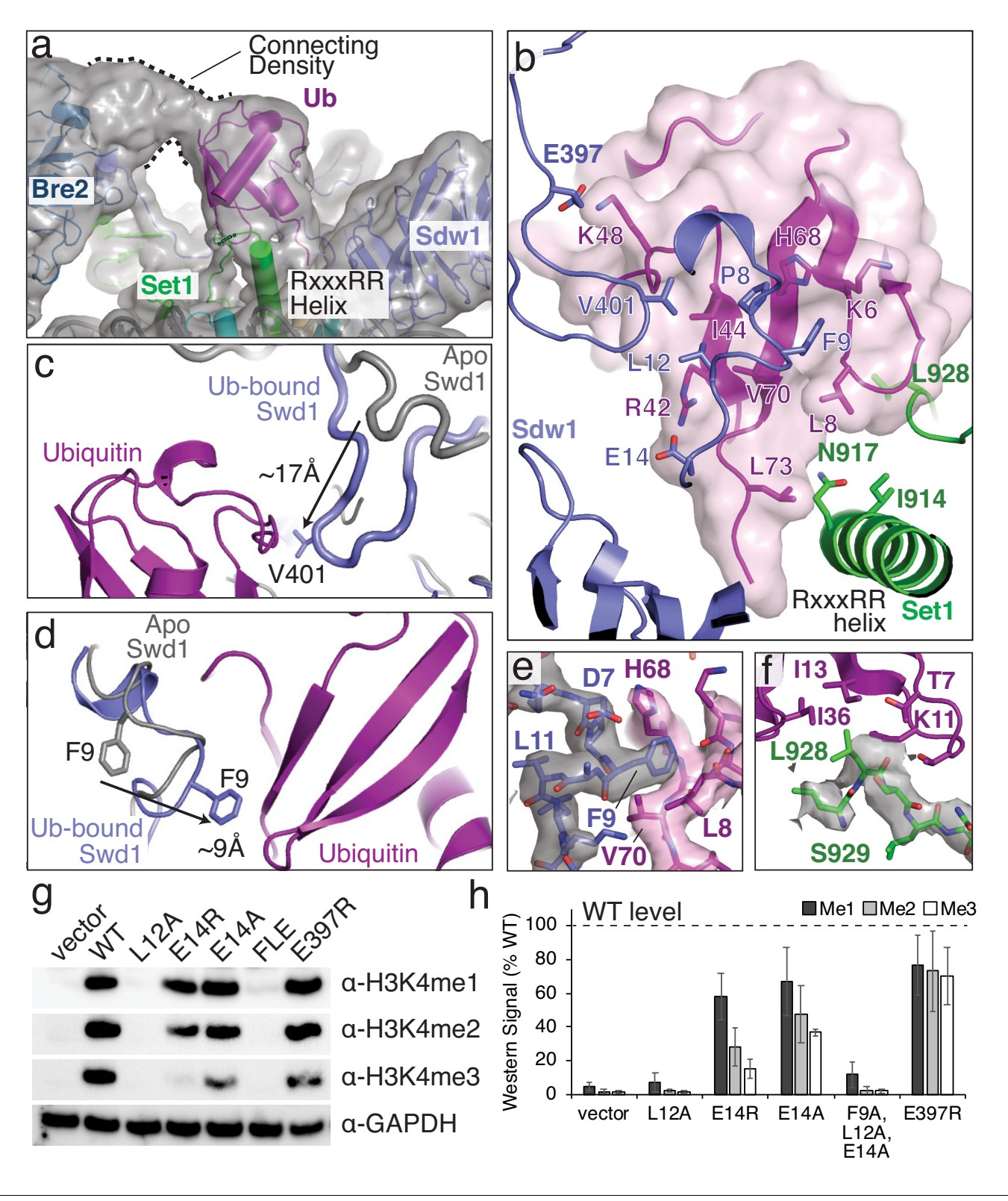

**Figure 6.** Structural basis for COMPASS recognition of H2B-ub. (a) Model of the COMPASS H2B-Ub nucleosome complex. The unsharpened EM density is shown as a semi-transparent gray surface. The connecting density between Bre2 and ubiquitin is shown. (b) Detailed view of the interactions made between Ubiquitin and Swd1 and Set1. Ubiquitin is enclosed in a semi-transparent pink surface and the interfacial residues are shown in stick representation. (c–d) Conformational changes in Swd1 that occur upon ubiquitin binding. The nucleosome-free (apo) structure of COMPASS is shown in

*Figure 6 continued on next page*

*Figure 6 continued*

gray (PDB: 6BX3). (**c**) Restructuring of the Swd1 C-terminal extension. (**d**) Restructuring of the Swd1 N-terminal extension. (**e**) Detailed view of Swd1 F9 interaction with ubiquitin. The sharpened EM density around Swd1 is shown as a semi-transparent gray surface and the EM density around ubiquitin is shown as a semi-transparent pink surface. (**f**) Detailed view of the interaction between Set1 L928 and ubiquitin. The sharpened EM density around Set1 is shown as a semi-transparent gray surface. (**g**) Western blot analysis of H3K4 methylation states in *swd1Δ* yeast strains transformed with plasmids containing the indicated *swd1* variants. (**h**) Quantification of the western blots from panel **g**. Error bars are the standard deviation of the data (n = 4). The online version of this article includes the following source data and figure supplement(s) for figure 6:

**Source data 1.** Western blot quantification for Swd1 Ubiquitin-binding mutants.
**Figure supplement 1.** Comparison of Set1 catalytic domains from COMPASS in different states.

interactions between Swd1 E14 and Ub R42 on one side, and between Swd1 E397 and Ub K48 on the other (*Figure 6b*).

In addition to the I44 patch contacts, Set1 contacts the hydrophobic 'I36 patch' of ubiquitin consisting of I36, L71 and L73 (*Figure 6b,f*) (*Komander and Rape, 2012*). While the I36 patch of ubiquitin is not as widely utilized by ubiquitin-binding proteins, it is also used by the Dot1L methyltransferase (*Figure 5c*). At the end of the RxxxRR helix Ub L73 contacts I914 and the aliphatic portion of N917 through van der Waals interactions (*Figure 6b*). The importance of the L73 contact is supported by previous in vitro studies showing that ubiquitin residues, L71 and L73, are critical for ubiquitin- dependent COMPASS activity (*Holt et al., 2015*). Finally, Set1 L928 inserts into a small hydrophobic pocket within ubiquitin and likely interacts with ubiquitin I13, I36, T7 and the aliphatic portion of K11 (*Figure 6f* and *Figure 1—figure supplement 4*). However, this region of the map does not show clear sidechain density for Set1 L928, so the position of the L928 sidechain is inferred from the conformation of the protein backbone.

To assess the contribution of interfacial Swd1 residues in ubiquitin-dependent H3K4 methylation in vivo, we generated *swd*1 deletion yeast strains expressing mutant Swd1 and measured the effects on H3K4 methylation. As shown in *Figure 6g–h*, the Swd1 L12A mutation completely abolished H3K4 methylation by COMPASS in vivo, indicating that the interaction between this Swd1 residue and Ub I44 is critical to COMPASS function. Swd1 E14A and E14R mutations both greatly decreased H3K4 di- and tri-methylation and also reduced H3K4 mono-methylation to an intermediate level (*Figure 6g,h*). Unsurprisingly, a Swd1 F9A, L12A and E14A triple mutant (FLE) completely abolished H3K4 methylation. Finally, the E397R mutation, which lies on the periphery of the ubiquitin interaction, modestly reduced all methylation states of H3K4 (*Figure 6g,h*). Together, these data show that the contacts observed between COMPASS and the H2B-linked ubiquitin are critical to activate COMPASS for H3K4 methylation.

## Discussion

Our structure reveals the basis of crosstalk between H2B-ubiquitination and H3K4 methylation by *Saccharomyces cerevisiae* COMPASS and shows that it methylates its target lysine in an asymmetric manner by recognizing the H2B-Ub and H3K4 on opposite sides of the nucleosome (*Figure 1d*). This asymmetric recognition is distinct from the H3K79 methyltransferase, Dot1L, which is also stimulated by H2B-Ub but which methylates H3 on the same, cis-H3, side of the nucleosome (*Figure 5c*) (*Anderson et al., 2019*; *Valencia-Sánchez et al., 2019*; *Worden et al., 2019*). To our knowledge, the asymmetric recognition of H2B-Ub and H3K4 by COMPASS and COMPASS-related complexes (*Hsu et al., 2019*; *Xue et al., 2019*) is the first example of trans-nucleosome histone crosstalk. This observation highlights the importance of asymmetry in histone modifications. Previous studies have shown that 'repressive' H3K27 tri-methyl and 'active' H3K4 tri-methyl marks can be deposited asymmetrically in the same nucleosome, but on opposite H3 tails (*Voigt et al., 2012*). This asymmetric, bivalent modification of H3K27 and H3K4 is believed to be associated with maintaining promoters in a poised state during differentiation. Our structure provides a mechanistic framework to understand how asymmetric nucleosome modifications can be read out and deposited during histone crosstalk.

Compared with other histone modifications such as methylation and acetylation, ubiquitination is a large, highly complex mark that presents a chemically rich interaction surface to potential binding partners and also decompacts chromatin (*Fierz et al., 2011*). In addition, because it is conjugated to the nucleosome through its flexible C-terminus, ubiquitin can adopt a range of orientations on the

nucleosome, enabling even greater complexity it its recognition. Several structures now exist of H2B-Ub-activated methyltransferases bound to ubiquitinated nucleosomes, which reveal highly divergent strategies for ubiquitin recognition (*Figure 5*). The distinct ubiquitin binding modes employed by COMPASS (*Hsu et al., 2019*) (this study), Dot1L (*Anderson et al., 2019*; *Valencia-Sánchez et al., 2019*; *Worden et al., 2019*) and the MLL1 core complex (*Xue et al., 2019*) reveal a striking plasticity in how ubiquitin is able to interact with and activate these different histone methyltransferases. The high complexity of the ubiquitin mark likely enables H2B-Ub to communicate with these different enzymatic complexes and template the deposition of 'activating' marks during transcription. The size of ubiquitin might also impose a significant steric obstacle that could inhibit the activity of enzymes which do not directly recognize the ubiquitin. Indeed, H2B-Ub may even impose a steric barrier to the transcriptional machinery as evidenced by the observation that efficient transcription elongation requires removal of H2B-Ub (*Wyce et al., 2007*).

Our structure suggests a mechanism by which H2B-Ub stimulates COMPASS to methylate H3K4. Compared to the nucleosome-free *Saccharomyces cerevisiae* COMPASS structure (*Qu et al., 2018*) and the structure of *Kluyveromyces lactis* COMPASS bound to an H3 peptide and SAM (*Hsu et al., 2018*), there are no discernable conformational changes in the Set1 catalytic domain that could explain how H2B-Ub stimulates methylation (*Figure 6—figure supplement 1*). It is notable that, for both COMPASS and Dot1L, the presence of ubiquitin conjugated to H2B does not appear to increase affinity of the enzyme for the nucleosome (*Figure 1—figure supplement 1*) (*Worden et al., 2019*). The lack of a discernable effect on binding energy suggests that ubiquitin binding primarily affects catalytic activation and, in the case of Dot1L, has been shown to increase $k_{cat}$ but not $K_M$ (*McGinty et al., 2009*; *Worden et al., 2019*). Alternatively, It has been suggested that ubiquitin activates Dot1L by using its binding energy to pay the energetic cost of inducing a conformational change in the globular core of histone H3, thereby inserting K79 into the Dot1L active site (*Worden et al., 2019*). We speculate that H2B-Ub may similarly activate COMPASS by providing the binding energy needed to induce the disordered Set1 RxxxRR motif to form a helix that mediates contacts with the nucleosome. Ubiquitin binding may also compensate for the energetic cost of inducing the Set1 helix, 326–331, to unravel and form an extended β-strand that buttresses the RxxxRR helix and the catalytic domain of Set1 (*Figure 4*). Our structure provides a basis for further elucidating the role that H2B ubiquitination plays in stimulating histone methyltransferase activity.

## Materials and methods

**Key resources table**

| Reagent type (species) or resource | Designation | Source or reference | Identifiers | Additional information |
|---|---|---|---|---|
| Strain, strain background (*Saccharomyces cerevisiae*) | Yeast strain BY4741 | (*Baker Brachmann et al., 1998*) | | |
| Antibody | Anti-Histone H3 (mono-methyl K4) polyclonal | Abcam | Abcam Cat# ab8895, RRID:AB_306847 | WB (1:500) |
| Antibody | Anti-Histone H3 (di-methyl K4) polyclonal | Abcam | Abcam Cat# ab7766, RRID:AB_2560996 | WB (1:500) |
| Antibody | Anti-Histone H3 (tri-methyl K4) polyclonal | Abcam | Abcam Cat# ab8580, RRID:AB_306649 | WB (1:500) |
| Antibody | Anti-GAPDH monoclonal | Abcam | Abcam Cat# ab125247, RRID:AB_11129118 | WB (1:2000) |
| Recombinant DNA reagent | pEW106 | This Study | | pQE-81L: H3 K4M |
| Recombinant DNA reagent | pEW66 | This Study | | COMPASS expression plasmid pBIG1a containing Bre2, Swd1, Swd2, Sdc1 and Sgh1 |

*Continued on next page*

Continued

| Reagent type (species) or resource | Designation | Source or reference | Identifiers | Additional information |
|---|---|---|---|---|
| Recombinant DNA reagent | pEW107 | This Study | | COMPASS expression plasmid pBIG1b containing 6xHis-3xFLAG-Set1(762–1080), Swd3 and twin-strep-Spp1. |
| Recombinant DNA reagent | pEW108 | This Study | | COMPASS expression plasmid pBIG1ab containing all COMPASS subunits. |
| Recombinant DNA reagent | WT Set1 | This Study | | pEW111 pRS415: WT Set1 |
| Recombinant DNA reagent | WT Swd1 | This Study | | pEW113 pRS415: WT Swd1 |
| Recombinant DNA reagent | Set1(R936A) | This Study | | pEW118 pRS415: Set1(R936A) |
| Recombinant DNA reagent | Set1(R936E) | This Study | | pEW120 pRS415: Set1(R936E) |
| Recombinant DNA reagent | Set1(R901A) | This Study | | pEW123 pRS415: Set1(R901A) |
| Recombinant DNA reagent | Set1(R904A) | This Study | | pEW124 pRS415: Set1(R904A) |
| Recombinant DNA reagent | Set1(R908A) | This Study | | pEW125 pRS415: Set1(R908A) |
| Recombinant DNA reagent | Set1(R909A) | This Study | | pEW126 pRS415: Set1(R909A) |
| Recombinant DNA reagent | Set1(R901A, R904A, R908A, R909A) | This Study | | pEW127 pRS415: Set1(R901A, R904A, R908A, R909A) |
| Recombinant DNA reagent | Set1(R901E) | This Study | | pEW128 pRS415: Set1(R901E) |
| Recombinant DNA reagent | Set1(R904E) | This Study | | pEW129 pRS415: Set1(R904E) |
| Recombinant DNA reagent | Set1(R908E) | This Study | | pEW130 pRS415: Set1(R908E) |
| Recombinant DNA reagent | Set1(R909E) | This Study | | pEW131 pRS415: Set1(R909E) |
| Recombinant DNA reagent | Set1(R901E, R904E, R908E, R909E) | This Study | | pEW132 pRS415: Set1(R901E, R904E, R908E, R909E) |
| Recombinant DNA reagent | Swd1(L12A) | This Study | | pEW139 pRS415: Swd1(L12A) |
| Recombinant DNA reagent | Swd1(E14R) | This Study | | pEW140 pRS415: Swd1(E14R) |
| Recombinant DNA reagent | Swd1(E14A) | This Study | | pEW141 pRS415: Swd1(E14A) |
| Recombinant DNA reagent | Swd1(F9A, L12A, E14A) | This Study | | pEW142 pRS415: Swd1(F9A, L12A, E14A) |
| Recombinant DNA reagent | Swd1(E397R) | This Study | | pEW144 pRS415: Swd1(E397R) |
| Recombinant DNA reagent | Swd1(V263A) | This Study | | pEW145 pRS415: Swd1(V263A) |
| Recombinant DNA reagent | Swd1(I264A) | This Study | | pEW146 pRS415: Swd1(I264A) |
| Recombinant DNA reagent | Swd1(N265A) | This Study | | pEW147 pRS415: Swd1(N265A) |

*Continued on next page*

*Continued*

| Reagent type (species) or resource | Designation | Source or reference | Identifiers | Additional information |
|---|---|---|---|---|
| Recombinant DNA reagent | Swd1(K266A) | This Study | | pEW148 pRS415: Swd1(K266A) |
| Commercial assay or kit | MTase-Glo | Promega | V7601 | |

## Expression and purification of COMPASS

Genes encoding subunits of COMPASS were cloned from yeast genomic DNA (S288C) using PCR amplification and inserted into pLIB vector from the biGBac baculovirus expression system (*Weissmann et al., 2016*) using Gibson assembly primers as described in the biGBac assembly protocol. Full-length Sgh1, Sdc1, Swd2, Swd1 and Bre2 were assembled into the pBIG1a expression vector (pEW66). Full-length Swd3, Spp1 containing two N-terminal strep tags (twin-strep-Spp1), and residues 762–1080 of Set1 with an N-terminal hexahistidine tag followed by three binding sites for the FLAG anitbody (6xHis-3xFLAG-Set1(762–1080)) were assembled into the pBIG1b expression vector (pEW107). The gene cassettes from the pBIG1a and pBIG1b expression vectors were excised using PmeI and cloned into pBIG2ab to form the final COMPASS expression assembly containing all eight COMPASS genes (pEW108). DH10Bac *E. coli* (Thermo-Fisher) was transformed with the pBIG2ab COMPASS expression vector to produce viral bacmids for baculovirus expression. SF9 insect cells were transfected with the COMPASS bacmid and the viral stock was amplified two times to obtain the high titer p3 viral stock. For protein production, 2L of 3 million cells/ml of High Five insect cells (Thermo-Fisher) were infected with the p3 viral stock at a MOI of 1. The insect cells were harvested after three days by centrifugation and then resuspended in 120 ml of lysis buffer (50 mM Tris pH 8.0, 200 mM NaCl, 10% glycerol) supplemented with 1 tablet of complete, EDTA-free protease inhibitor (Roche) per 50 ml of buffer. The resuspended tablet was flash frozen in liquid nitrogen and stored at −80˚C.

To purify the complex, the cell pellet was thawed at room temperature and then placed on ice. All subsequent purification steps were conducted at 0–4˚C. The thawed cell pellet was diluted to 150 ml using lysis buffer and lysed with an LM10 Microfluidizer (Microfluidics). The lysate was clarified by centrifugation and then filtered through a 0.45 μm fast-flow bottle-top filter (Nalgene). The filtered lysate was batch-adsorbed to 5 ml of freshly equilibrated M2-FLAG affinity gel (Millipore Sigma) and incubated for 1 hr. The sample was washed in batch with 40 ml of lysis buffer, transferred to a gravity flow column and washed with 20 ml of lysis buffer. COMPASS was then eluted from the FLAG resin using lysis buffer supplemented with 0.15 mg/ml 3x FLAG peptide. 5 mM β-mercapto-ethanol (BME) was added to the eluate and immediately poured onto a column containing with Streptactin-XT resin (IBA). The column was washed with five column volumes of lysis buffer and the complex eluted with lysis buffer supplemented with 50 mM Biotin. The strep eluate was concentrated to 0.5 ml and then injected onto a Superose six size exclusion column (GE) equilibrated with GF buffer (30 mM HEPES pH 7.8, 150 mM NaCl, 1 mM TCEP, 5% glycerol). The following day, peak fractions were pooled, concentrated, flash frozen in liquid nitrogen and stored at −80˚C.

## Purification of histone proteins

Unmodified *Xenopus laevis* histone proteins, ubiquitinated histone H2B and histone H3 containing norleucine in place of H3K4 were purified as described previously (*Dyer et al., 2004*; *Worden et al., 2019*). In brief, ubiquitinated H2B-K120Ub was prepared using H2B containing a K120C substitution and ubiquitin containing a G76C substitution. Ubiquitin was cross-linked to H2B-K120C using dichloroacetone (DCA) as described (*Morgan et al., 2016*). Histone H3 with K4 substituted with norleucine was generated by mutating K4 to methionine (pEW106) and expressing the H3K4M protein in minimal medium supplemented with all amino acids except methionine, as well as with added norleucine (*Worden et al., 2019*).

## Purification of 601 DNA

The pST55−16 × 601 plasmid was a generous gift from Dr. Song Tan (*Makde et al., 2010*). The pST55−16 × 601 plasmid containing 16 repeats of the 147 base pair Widom 601 positioning sequence (*Lowary and Widom, 1998*) was grown in the *E. coli* stain XL-1 blue. The plasmid was purified and the 601 DNA was excised with *Eco*RV and recovered essentially as described previously (*Dyer et al., 2004*).

## Nucleosome reconstitution

Nucleosomes were prepared as previously described (*Dyer et al., 2004*).

## Electrophoretic Mobility Shift assays

For the electrophoretic mobility shift assays (EMSA), 50 nM of each nucleosome variant was mixed with COMPASS at the indicated concentrations and incubated at room temperature for 30 min in EMSA buffer (20 mM HEPES pH 7.5, 100 mM NaCl, 1 mM DTT, 0.2 mg/ml bovine serum albumin (BSA), 100 μM S-adenosyl methionine (SAM)). Samples were then diluted with an equal volume of 2x EMSA sample buffer (40 mM HEPES pH 7.6, 100 mM NaCl, 10% Sucrose, 2 mM DTT, 0.2 mg/ml BSA) and 10 μl of sample was loaded onto a 6% native Tris-borate EDTA (TBE) gel at 4°C. The gel was stained with SybrGold DNA stain (Thermo-Fisher) to visualize bands.

## Yeast strains

All yeast strains were prepared from the BY4743 background and cultured using standard methods. The Swd1Δ strain was obtained from the Yeast Knock-Out collection (Dharmacon) and was a generous gift from Dr. Carol Greider. The Set1Δ strain was prepared using PCR-mediated gene disruption with the KanMX gene as a selectable marker. Wild type (WT) SWD1 and SET1 were isolated by PCR from *Saccharomyces cerevisiae* genomic DNA with a native 600 base pair (bp) upstream promotor and a 200 bp terminator. The isolated SWD1 and SET1 genes were cloned into pRS415 and mutants were generated using inverse PCR. Empty pRS415 vector or plasmids containing WT or mutant variants of SWD1 or SET1 were introduced to the Swd1Δ or Set1Δ strains using standard yeast transformation techniques and Leucine selection.

## Protein extraction and western blot analysis

Yeast deletion strains containing the pRS415 vector, WT or mutant variants of Swd1 or Set1 were grown in 50 ml of SD-Leu media at 30°C to an $OD_{600}$ of 0.7–1.0. A volume of 40 ml of the yeast culture was pelleted and resuspended in 10% tri-chloroacetic acid (TCA) to a final volume with an $OD_{600}$ of 6 and incubated at room temperature for 30 min. 3 ODs of cells (500 μl) were aliquoted into 1.7 ml Eppendorf tubes, pelleted and frozen. For total protein extraction, the cell pellet was thawed on ice and resuspended in 250 μl of 20% TCA. A volume of 250 μl of 0.25 mm - 0.5 mm glass beads were added to the resuspended cell pellet and the cells were lysed by vortexing for 6 min. The bottom of the Eppendorf tube was punctured, placed into a fresh tube, and the lysed cells were collected by centrifugation. The glass beads were washed with an additional 300 μl of 5% TCA and discarded. Total protein was pelleted by centrifugation at 20,000 G for 10 min at 4°C. The resulting pellet was washed with 100% ethanol at −20°C and resuspended in 2x SDS sample buffer. The efficiency of the total protein extraction was evaluated by SDS PAGE followed by stain-free protein imaging (BioRad). For Western blotting, equal amounts of protein extraction were separated by SDS-PAGE, transferred to PVDF membranes, blocked with 5% milk in TBST buffer and probed with anti α-H3me1 (Abcam Cat# ab8895, RRID:AB_306847) and anti α-H3me2 (Abcam Cat# ab7766, RRID:AB_2560996) and anti α-H3me3 (Abcam Cat# ab8580, RRID:AB_306649) and anti α-GAPDH (Abcam Cat# ab125247, RRID:AB_11129118) antibodies. A total of three technical replicates were analyzed from the same yeast growth.

## COMPASS activity assay

A 2x concentrated 2-fold dilution series of COMPASS (2560 μM – 40 nM) was prepared in reaction buffer (20 mM HEPES pH 7.5, 100 mM NaCl, 1 mM DTT, 0.2 mg/ml BSA). To initiate the reaction, 6 ul of each 2x stock of COMPASS was added to 6 ul of 500 μM H3 peptide (residues 1–21) dissolved in reaction buffer that either contained 500 uM SAM or contained no SAM. The reaction was allowed

progress at 25°C for 35 min and then the reaction was quenched by the addition of 3 ul 0.5% tri-fluoroacetic acid (TFA). 10 ul of the quenched reaction was transferred to a microplate and the amount of S-adenosyl homocysteine (SAH) was quantified using the MTase-Glo assay (Promega) according to the manufacture's instructions. Raw luminescence values were measured on a POLAR-star Omega fluorescence plate reader (BMG Labtech).

## Cryo EM sample preparation

A 2.44 ml volume of 300 nM COMPASS, 100 nM of nucleosome containing H2B-Ub and H3K4Nle, and 200 µM SAM was prepared in crosslinking buffer (25 mM HEPES pH7.5, 100 mM NaCl, 1 mM DTT). The sample was incubated on ice for 30 min and then mixed with 2.44 ml of 0.14% glutaralde-hyde to initiate crosslinking. The crosslinking reaction progressed on ice and was quenched after one hour by the addition of 1 M Tris pH 7.5 to a final concentration of 100 mM. The reaction was incubated for 1 hr on ice and then concentrated to ~50 µl using an Amicon Ultra 30K MWCO spin concentrator. The concentration of the complex was determined using the absorption at 260 nm of the 601 nucleosome DNA. Quantifoil R2/2 grids were glow discharged for 45 s at 15 mA using a Pelco Easyglow glow discharger. A volume of 3 µl of 0.5 mg/ml crosslinked sample was added to the glow-discharged grids and flash frozen in liquid ethane using a Vitrobot (Thermo Fisher) at 4°C and 100% humidity with a 3.5 s blot time.

## EM data collection and refinement

All data were collected at the National Cryo-Electron Microscopy Facility (NCEF) at the National Cancer Institute on a Titan Krios (Thermo-Fisher) at 300 kV utilizing a K3 (Gatan) direct electron detector in counting-mode with at a nominal magnification of 81,000 and a pixel size of 1.08 Å. Data were collected at a nominal dose of 50 e⁻/Å$^2$ with 40 frames per movie and 1.25 e⁻/frame. A total of 5784 movies were collected.

The dataset was processed in Relion 3.0 (*Zivanov et al., 2018*). All movies were motion-corrected and dose-weighted using the Relion 3.0 implementation of MotionCorr2 (*Zivanov et al., 2018*). An initial batch of 500 micrographs were randomly selected and used to pick 315,372 particles using the Laplacian of gaussian auto-picking feature. After 3 rounds of 2D-classification to remove junk particles, 216,107 particles were used for 3D classification. A single good class of 79,832 particles was refined and used as a model for template-based picking on the entire dataset, resulting in 2,036,654 particles. The particles were extracted and binned by a factor of 4. 1,357,004 particles were retained after 2D classification and used for 3D classification with six classes. Three classes emerged from the 3D classification which seemed to have a well resolved COMPASS on at least one side of the nucleosome. These three classes (1103264 particles) were merged and subjected to another round of 3D classification with four classes using a mask that encompassed the nucleosome and one COMPASS molecule. Two of the resulting Classes appeared to have high resolution features, were merged (650847 particles) and subjected to 3D refinement using the same mask that encompassed the nucleosome and COMPASS. After refinement the particles were reextracted at full resolution and refined again using the same mask. The unbinned particle stack was subjected beam tilt correction and per-particle contrast transfer function (CTF) estimation in Relion 3.0. The final particle stack was then subjected to masked refinement using the same mask that was used in the previous masked refinement and classification steps. The final structure was sharpened using the Relion postprocessing tool with a soft mask that encompassed the more well resolved COMPASS molecule and the nucleosome. A sharpening B-factor of −122.6 Å$^2$ was applied. The final resolution of the COMPASS-nucleosome structure is 3.37 Å according to the Fourier shell correlation (FSC) 0.143 criterion.

## Model building and refinement

Coordinates for the *Xenopus laevis* nucleosome core particle (PBD: 6NJ9), *Saccharomyces cerevisiae* Spp1, and the N-set domain of Set1 (PDB: 6 × 3) were docked into the EM density using Chimera. Crystal structures of *Kluyveromyces lactis* Sdc1, Swd3, Swd1, Bre2 and the Set1 catalytic domain (PDB: 6CHG) fit the EM density better than the existing cryo-electron microscopy structures (PDB: 6 × 3) of *S. cerevisiae* Sdc1 (25% identity, 67% in modeled area), Swd3 (50% identity), Swd1 (50% identity), Bre2 (42% Identity) and Set1 (41% identity, 86% in modeled

area). Therefore, homology models of *K. lactis* Sdc1, Swd3, Swd1, Bre2 and the Set1 catalytic domain were prepared using the Swiss-model software and docked into the EM density using Chimera. The resulting model was iteratively refined in Phenix (*Afonine et al., 2018*) using reference restraints for Ubiquitin, the Set1 N-set domain, Spp1, Bre2 and the Sdc1 dimer (from PDBs 1UBQ, 6B × 3 and 6CHG) and edited in COOT (*Emsley et al., 2010*). The model was refined against the full map and any overfitting of the model was assessed by calculating the model-map FSC between the refined model and the sharpened, masked half-maps (Half one and Half 2) that were filtered to the FSC = 0.143 resolution cutoff for each half map (3.96 Å). The model-map FSCs of the two half maps agree well indicating that there is little overfitting of the model. Furthermore, the FSC = 0.5 resolution estimate of the model/map (Full) does not exceed the calculated map resolution, indicating that the model is not overfit.

## Acknowledgements

This research was supported, in part, by the National Cancer Institute's National Cryo-EM Facility at the Frederick National Laboratory for Cancer Research under contract HSSN261200800001E. EJW is a Damon Runyon Fellow supported by the Damon Runyon Cancer Research Foundation (DRG 2308–17). This work was supported by grant GM130393 from the National Institute of General Medical Sciences.

## Additional information

### Competing interests

Cynthia Wolberger: Senior editor, *eLife*. The other authors declare that no competing interests exist.

### Funding

| Funder | Grant reference number | Author |
| --- | --- | --- |
| National Institute of General Medical Sciences | GM130393 | Cynthia Wolberger |
| Damon Runyon Cancer Research Foundation | DRG 2308-17 | Evan J Worden |

The funders had no role in study design, data collection and interpretation, or the decision to submit the work for publication.

### Author contributions

Evan J Worden, Conceptualization, Data curation, Formal analysis, Validation, Investigation, Visualization, Methodology, Writing - original draft, Writing - review and editing, Isolated COMPASS genes, Purified proteins, Prepared cryo-EM samples, Screened cryo-EM conditions, Determined the structure, Conducted the yeast experiments and cloned COMPASS mutants; Xiangbin Zhang, Investigation, Methodology, Cloned the COMPASS expression constructs, Expressed COMPASS and cloned COMPASS mutants; Cynthia Wolberger, Conceptualization, Supervision, Funding acquisition, Writing - original draft, Writing - review and editing

### Author ORCIDs

Evan J Worden (iD) https://orcid.org/0000-0003-0644-166X
Cynthia Wolberger (iD) https://orcid.org/0000-0001-8578-2969

### Decision letter and Author response

Decision letter https://doi.org/10.7554/eLife.53199.sa1
Author response https://doi.org/10.7554/eLife.53199.sa2

## Additional files

### Supplementary files

• Transparent reporting form

### Data availability

Coordinates have been deposited in the PDB under accession code 6VEN. Maps have been deposited in EMDB under accession codes EMD21157.

The following datasets were generated:

| Author(s) | Year | Dataset title | Dataset URL | Database and Identifier |
|---|---|---|---|---|
| Worden EJ, Wolberger C | 2020 | Yeast COMPASS in complex with a ubiquitinated nucleosome | https://www.rcsb.org/structure/6VEN | RCSB Protein Data Bank, 6VEN |
| Worden EJ, Wolberger C | 2020 | Yeast COMPASS in complex with a ubiquitinated nucleosome | https://www.ebi.ac.uk/pdbe/entry/emdb/EMD-21157 | Electron Microscopy Data Bank, EMD-21157 |

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
