## [Decision Letter]

**Acceptance summary:**

As noted in the initial review, Worden et al. have written a beautiful paper describing the structure and the underlying mechanism by which the COMPASS complex recognizes the H2B-ubiquitinated nucleosome. During the revision, the authors improved the resolution of the cryo-EM structure of a complex of *S. cerevisiae* COMPASS with H2B-Ub nucleosome. The structure reveals the overall organization of the complex, including the mechanism of recognizing ubiquitin and selection of the trans H3 histone K4 for modification. As we noted earlier, what makes this paper have high potential impact is not only the structural information, but the in vivo work that complements the structure. Overall, the experiments are of high technical quality.

**Decision letter after peer review:**

Thank you for submitting your article "Structural basis for COMPASS recognition of an H2B-ubiquitinated nucleosome" for consideration by *eLife*. Your article has been very favorably reviewed by three peer reviewers, and the evaluation has been overseen by John Kuriyan as the Reviewing and Senior Editor. The following individuals involved in review of your submission have agreed to reveal their identity: Christopher P Hill (Reviewer #1); Karolin Luger (Reviewer #3).

The reviewers have discussed the reviews with one another and the Reviewing Editor has drafted this decision to help you prepare a revised submission.

Review:

Worden et al. have written a beautiful paper describing the structure and the underlying mechanism by which the COMPASS complex recognizes the H2B-ubiquitinated nucleosome. The cryo-EM structure of a complex of *S. cerevisiae* COMPASS with H2B-Ub nucleosome is reported at about 4Å resolution. The structure reveals the overall organization of the complex, including the mechanism of recognizing ubiquitin and selection of the trans H3 histone K4 for modification. What makes this paper such high potential impact is not only the structural information, but the in vivo work that complements the structure. Overall, the experiments are of high technical quality.

Previous genetic data and biochemical data here strongly validate the structural model presented. The model certainly helps understand, but does not completely explain, the requirement H2B-Ub for K4 methylation. Interestingly, it's shown that H2B-Ub does not increase COMPASS-nucleosome affinity or alter the catalytic site, but it does induce some rearrangements in regions of Set1 around the active site. A paper in press at Molecular Cell (here cited as a bioRxiv preprint from Hsu et al.) presents a very similar structure for the *K. lactis* COMPASS, so this paper is very timely and should be published rapidly.

The reviewers have raised several questions or concerns. We are not asking for additional experimental work to be done, but please address these points in a revised manuscript.

Nomenclature:

1) Please note that the "CPS" gene names used by Shilatifard are not the standard names agreed to and commonly used by the yeast community (www.yeastgenome.org). I would encourage the authors to switch to the standard names, but at the very least they need to include the translations (i.e. Cps60/Bre2, Cps50/Swd1, etc) when first introducing the proteins.

2) People interested in this paper will also read the Hsu paper and other structural studies of the Set1/MLL complexes. Many of those papers refer to the RXR as the ARM (Arginine Rich Motif), so it would be good to mention that early on to help readers make comparisons.

3) Introduction paragraph three. Please define the difference between "core" and "H2B-Ub-sensing" subcomplexes, as this isn't obvious, even to somebody reasonably familiar with COMPASS.

4) Subsection “COMPASS interacts with DNA using three distinct interfaces”, last sentence in the first paragraph: "Hotspot" is probably not the appropriate term here.

Other points:

5) A recent paper from Chaudhury et al. (G+D 2019) presented evidence suggesting dimerization of COMPASS, with the two complexes enclosing the nucleosome. Their model looks incompatible with the data presented here. Could the authors comment on this here or in the Discussion where they discuss asymmetrical modifications?

6) Results section, Figures 2, 3, 6. At several points in the paper, the authors propose specific inter-subunit side chain interactions based on what appears (in the figures) to be weak density data, and some of the predicted mutants in these residues have very little effect. The overall resolution of the cryo-EM data is not that high, and much of the model is based on docking in other structures (including from another species), so one needs to be careful about overstating conclusions. Perhaps the authors could make clear which contacts are unambiguous and which are more speculation.

7) Given the other paper in press on the *K. lactis* COMPASS, there were a few lines that didn't seem quite right. In the Introduction, it's strange to say there is "currently no structural information" about COMPASS/H2B-Ub when you cite the Hsu 2019 pre-print in other parts of the paper. The Discussion section should also reference the new Hsu 2019 paper along with "this study". The two papers are nicely complementary and contemporaneous, so the authors might as well be generous in giving credit where it's due.

8) Are data available to demonstrate that the COMPASS complex used in this study is catalytically active? It would be good to know the catalytic efficiency in methylation. While authors show that COMPASS can bind to both H2B-Ub and unmodified H2B nucleosomes, they do not reveal if both can activate. This is interesting because, as seen in Xue et al., 2019, the human MLL1 complex is stabilized by the H2B-Ub nucleosome in a favorable conformation that promotes H3K4 methylation. It might be worth doing biochemical assays to see how this ubiquitination is conserved amongst paralogs, and to what extent it promotes activity.

9) While the western blots are discussed well throughout the paper, there are concerns with quantification. The authors discuss how mutations lead to decrease of H3K4 methylation patterns, but nowhere are the western blots quantified (with error bars) to show how much difference there is between mutants (and replicates). Therefore, western blots should be quantified. Furthermore, these western blots should have a control (such as GAPDH) in all lanes to indicate differences in mutants is not simply a gel-loading difference. In addition to quantifying western blots, the EMSA done in Figure 1—figure supplement 1 comparing nucleosome binding should also be quantified and fit to a binding-curve to provide better comparisons.

10) It is interesting that COMPASS can do H3K4 mono-, di-, and tri-methylation, while the human paralog MLL3 primarily does mono-methylation and MLL1 primarily does di- and tri-methylation. In agreement with the importance of this, certain mutations from this study show differences in methylation states, indicating the mechanism of mono-, di-, and tri-methylation may have some underlying differences. The differences in methylation states should be discussed for the mutants.

11) Please provide more information on DNA used (both sequence and length).

---

## [Author Response]

The reviewers have raised several questions or concerns. We are not asking for additional experimental work to be done, but please address these points in a revised manuscript.Nomenclature:1) Please note that the "CPS" gene names used by Shilatifard are not the standard names agreed to and commonly used by the yeast community (www.yeastgenome.org). I would encourage the authors to switch to the standard names, but at the very least they need to include the translations (i.e. Cps60/Bre2, Cps50/Swd1, etc) when first introducing the proteins.

We thank the reviewers for their suggestion on nomenclature for the individual COMPASS subunits. We have switched the naming convention used in the manuscript to the standard which is most commonly used in the yeast community. We have also included citations to Roguev et al. and to Nagy et al. in the Introduction which concurrently reported the discovery of COMPASS and propose their own independent naming conventions.

“Cps15 (Sgh1), Cps25 (Sdc1), Cps30 (Swd3), Cps35 (Swd2), Cps40 (Spp1), Cps50 (Swd1), Cps60 (Bre2), as well as Set1 (Miller et al., 2001; Nagy et al., 2002; Roguev et al., 2001; Shilatifard, 2012)”

2) People interested in this paper will also read the Hsu paper and other structural studies of the Set1/MLL complexes. Many of those papers refer to the RXR as the ARM (Arginine Rich Motif), so it would be good to mention that early on to help readers make comparisons.

We thank the reviewers for making this helpful suggestion. To aid in comparison with our structure we have added text which explicitly state that the RxxxRR motif is also referred to as the Arginine Rich Motif. To aid with clarity, we have additionally removed the shorthand “RxR” and refer the motif using its full name (“RxxxRR”) throughout the paper. We have also included a citation to Hsu et al.

“RXXXRR (RxR) motif in the N-set region of Set1 which is also referred to as the Arginine Rich Motif (ARM) (Hsu et al., 2019; Kim et al., 2013).”

3) Introduction paragraph three. Please define the difference between "core" and "H2B-Ub-sensing" subcomplexes, as this isn't obvious, even to somebody reasonably familiar with COMPASS.

We have added a sentence at the end of the paragraph that further explains the distinction between the two subcomplexes of COMPASS.

“We refer to this core subcomplex plus Spp1 as the H2B-ubiquitin-sensing subcomplex.”

4) Subsection “COMPASS interacts with DNA using three distinct interfaces”, last sentence in the first paragraph: "Hotspot" is probably not the appropriate term here.

We have changed “hotspot” to “highly utilized surface”, which better represents the function of the Swd1 loop.

“and the core histone octamer, making this part of Swd1 a highly utilized surface for COMPASS interaction with the nucleosome.”

Other points:5) A recent paper from Chaudhury et al. (G+D 2019) presented evidence suggesting dimerization of COMPASS, with the two complexes enclosing the nucleosome. Their model looks incompatible with the data presented here. Could the authors comment on this here or in the Discussion where they discuss asymmetrical modifications?

We thank the reviewers for pointing out this interesting paper and agree that their model is incompatible with our structure. We note that we do not find the data definitive regarding whether COMPASS dimerizes and whether the lower molecular weight complex observed in the sdc1 deletions and mutants indeed represents a COMPASS monomer. Neither this manuscript nor previous citations in the support of COMPASS dimerization include experiments such as sec-mals, mass spec or centrifugation that would be needed to determine the composition of the complexes that migrate on a size exclusion column with an apparent molecular weight of 1MDa or 400kDa, which are identified as dimeric or monomeric COMPASS, respectively. This then raises questions on how to interpret the observed differences in H3K4 methylation patterns observed in vivo. We feel that most of the effects shown for the Sdc1 deletion and mutation studies could be explained by the overall lower activity of COMPASS in these mutants (see Figure 2 in the Chaudhury paper). Rather than bringing up questions about the validity of this paper we prefer not to comment on the results as we have many issues regarding the authors interpretation of their data.

6) Results section, Figures 2, 3, 6. At several points in the paper, the authors propose specific inter-subunit side chain interactions based on what appears (in the figures) to be weak density data, and some of the predicted mutants in these residues have very little effect. The overall resolution of the cryo-EM data is not that high, and much of the model is based on docking in other structures (including from another species), so one needs to be careful about overstating conclusions. Perhaps the authors could make clear which contacts are unambiguous and which are more speculation.

During the review process, we undertook an alternate EM processing pipeline with the hopes of increasing the number of particles in the final model and also increase the map resolution. We are excited to report that our new approach increased the average resolution of the map from 3.9 Å to 3.37 Å. This increase in resolution also resulted in a more featured map, so many contacts that were not supported by strong sidechain density now have clear sidechain density or sidechain bumps that support our conclusions. We have updated all the figures with the new map and model and have prepared a new supplemental figure (Figure 1—figure supplement 4) that gives more detailed views and density for the reported contact surfaces. We have also updated the text to specifically state potential contacts that are not supported by clear density.

“We note that we did not observe clear sidechain density for Bre2 K318, so the position of this sidechain is inferred from the conformation of the protein backbone.”

“We note, however, that there is no clear sidechain density for R909 in our structure, so the contact between R909 and S112 is inferred from the conformation of the backbone”

“However, this region of the map does not show clear sidechain density for Set1 L928, so the position of the L928 sidechain is inferred from the conformation of the protein backbone.”

7) Given the other paper in press on the K. lactis COMPASS, there were a few lines that didn't seem quite right. In the Introduction, it's strange to say there is "currently no structural information" about COMPASS/H2B-Ub when you cite the Hsu 2019 pre-print in other parts of the paper. The Discussion section should also reference the new Hsu 2019 paper along with "this study". The two papers are nicely complementary and contemporaneous, so the authors might as well be generous in giving credit where it's due.

We thank the reviewers for pointing out this oversight. We have updated the text to include a reference to the recent Mol Cell paper by Hsu et al. and have also cited that paper.

“In addition, a recent structure of the related COMPASS complex from *K. lactis* has shown how COMPASS binds to a ubiquitinated nucleosome (Hsu et al., 2019). However, there is currently no structural information on how the full H2B-ubiquitin-sensing COMPASS subcomplex from *Saccharomyces cerevisiae* binds and recognizes the H2B-Ub containing nucleosome.”

8) Are data available to demonstrate that the COMPASS complex used in this study is catalytically active? It would be good to know the catalytic efficiency in methylation. While authors show that COMPASS can bind to both H2B-Ub and unmodified H2B nucleosomes, they do not reveal if both can activate. This is interesting because, as seen in Xue et al. 2019, the human MLL1 complex is stabilized by the H2B-Ub nucleosome in a favorable conformation that promotes H3K4 methylation. It might be worth doing biochemical assays to see how this ubiquitination is conserved amongst paralogs, and to what extent it promotes activity.

We tested the activity of COMPASS on an H3 peptide to confirm that the complex is catalytically active and have now included this data in Figure 1—figure supplement 1. We have also included a new sentence in the Results first paragraph that describes the activity of COMPASS on the H3 peptide. We also note that there is substantial supporting literature that clearly shows the native COMPASS complex, and also the specific subcomplex used in our study, is stimulated by H2B-ub both in vivo and in vitro (Dover et al., 2002; Hsu et al., 2019; Kim et al., 2013; Sun and Allis, 2002). Since it has been shown previously that this specific subcomplex of COMPASS is robustly stimulated by H2B, it is our opinion that showing this again would not add to the conclusions of the paper.

In relation to the comparison with MLL, we feel that an entire biochemical characterization of the differences between the Set1 and MLL complexes on H2B-Ub stimulation would be beyond the scope of this study.

9) While the western blots are discussed well throughout the paper, there are concerns with quantification. The authors discuss how mutations lead to decrease of H3K4 methylation patterns, but nowhere are the western blots quantified (with error bars) to show how much difference there is between mutants (and replicates). Therefore, western blots should be quantified. Furthermore, these western blots should have a control (such as GAPDH) in all lanes to indicate differences in mutants is not simply a gel-loading difference. In addition to quantifying western blots, the EMSA done in Figure 1—figure supplement 1 comparing nucleosome binding should also be quantified and fit to a binding-curve to provide better comparisons.

We have repeated the western blot experiments on the mutant yeast strains with a GAPDH loading control as requested. In addition, we quantified the intensity of the western blot signal for each replicate and have included bar graphs in Figures 3, 4 and 6 that show the average intensity of the western signal for each sample as a percent of WT. This analysis has not changed any of the conclusions from our manuscript.

We attempted to quantify the binding affinity of COMPASS to the unmodified, H2B-Ub and H2B-Ub + H3K4Nle nucleosomes, however only the Nle-containing nucleosomes produced a curve that could be fit to a binding isotherm. This is because binding of the unmodified and H2B-Ub nucleosome samples appears to change linearly over the concentration range tested and could therefore not be fit. We have included our binding analysis in Figure 1—figure supplement 1 and also added text that clarifies how to compare the different EMSAs in Figure 1—figure supplement 1.

“However, H2B-Ub nucleosomes that also contain the H3K4Nle mutant bind COMPASS with 2-5 fold higher affinity than unmodified nucleosomes (Figure 1—figure supplement 1, compare the 0.125 µM lane for all samples)”

10) It is interesting that COMPASS can do H3K4 mono-, di-, and tri-methylation, while the human paralog MLL3 primarily does mono-methylation and MLL1 primarily does di- and tri-methylation. In agreement with the importance of this, certain mutations from this study show differences in methylation states, indicating the mechanism of mono-, di-, and tri-methylation may have some underlying differences. The differences in methylation states should be discussed for the mutants.

We agree that the different product specificities between the Set1 COMPASS complexes and the MLL complexes are very important. However, because our western blot analyses are simply reading out the steady-state levels of H3K4 methylation in vivo, it is impossible to determine if the differences we observe are due to altered product specificities of COMPASS or simply slower enzyme turnover. Given that Set1/COMPASS is probably a distributive enzyme, like MLL1 and Dot1L (Frederiks et al., 2008; Patel et al., 2009), a slower enzyme turnover would result in an altered distribution of methylation states. To determine if the mutations are actually changing the product specificity of COMPASS, we would need to purify the mutant complexes and measure their product preferences in vitro, which we feel is beyond the scope of our study. To address the differences we see in the distribution of H3K4 methylation for our COMPASS mutants, we have added a sentence which points out that the actual cause of the change in methylation is uncertain.

“We note that it is not possible to determine from these data if the changes in the relative H3K4 mono-, di- and tri-methylation levels in the COMPASS mutants are caused by altered product specificity, or slower enzyme turnover.”

11) Please provide more information on DNA used (both sequence and length).

We have added a section to the Materials and methods describing the DNA used in the nucleosome reconstitution.

“Purification of 601 DNA

The pST55-16x601 plasmid was a generous gift from Dr. Song Tan (Makde et al., 2010). The pST55-16x601 plasmid containing 16 repeats of the 147 base pair Widom 601 positioning sequence (Lowary and Widom, 1998) was grown in the *E. coli* stain XL-1 blue. The plasmid was purified and the 601 DNA was excised with *Eco*RV and recovered essentially as described previously (Dyer et al., 2004).”

**References**

Frederiks, F., Tzouros, M., Oudgenoeg, G., van Welsem, T., Fornerod, M., Krijgsveld, J., and van Leeuwen, F. (2008). Nonprocessive methylation by Dot1 leads to functional redundancy of histone H3K79 methylation states. Nature structural & molecular biology *15*, 550-557.